# FABRIKx: Tackling the Inverse Kinematics Problem of Continuum Robots with Variable Curvature

**Dmitrii Kolpashchikov [1],\*, Olga Gerget [2] and Viacheslav Danilov [3,4]**

1   School of Computer Science and Robotics, Tomsk Polytechnic University, 634000 Tomsk, Russia
2   Institute of Control Sciences, 117997 Moscow, Russia
3   Quantori, Cambridge, MA 02142, USA
4   Department of Mechanical Engineering, Politecnico di Milano, 20133 Milan, Italy
*   Correspondence: dyk1@tpu.ru

**Abstract:** A continuum robot is a unique type of robots which move because of the elastic deformation of their bodies. The kinematics of such robots is typically described using constant curvature assumption. Such an assumption, however, does not completely describe the kinematics of a real-life continuum robot. As a result, variable curvature assumptions describe the kinematics of the continuum robot better, however, they are more complicated to formulate and work with. In particular, the existing methods of solving the inverse kinematics problem of multisection continuum robots with variable curvature suffer from a variety of deficiencies. Those deficiencies include complex matrix calculations, singularity problems, unscalability, and inability to find a numeric solution in some cases. In this work, we present FABRIKx: fast and reliable algorithm to solve the problem of inverse kinematics of the multisection continuum robot with variable curvature. In particular, to describe the variable curvature, we utilize a piecewise constant curvature assumption. The proposed algorithm combines both tangent and chord approaches to solve the inverse kinematics problem. The inverse kinematics of a single bending section of piecewise constant curvature is also described. To evaluate FABRIKx effectiveness, we compare it with the Jacobian-based and FABRIKc-based algorithms via simulation studies for different robots. The obtained results show that FABRIKx demonstrates a higher success rate and a lower solution time.

**Keywords:** continuum robots; inverse kinematics; forward kinematics; FABRIK

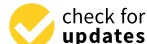



## 1. Introduction

Continuum robots are flexible manipulators that move because of the elastic deformation of their bodies. The flexible design of continuum robots makes them able to avoid unwanted collisions in a confined workspace, navigate to hard-to-access areas, and grasp objects using their body. They are used for different purposes including machining [1], non-destructive testing, and repairs inside complex devices [2–4]. In addition, continuum robots are able to reach hard-to-access and dangerous areas, such as outer space [5,6] and underwater [7]. Another sphere actively using these devices is medicine, where the continuum robots are used as endoscopes and surgical instruments for minimally invasive surgery [8–10].

Inverse kinematics is crucial for providing autonomy, motion planning, trajectory optimization, and obstacle avoidance for continuum robots. Despite significant attention from researchers, inverse kinematics for multisection continuum robots remains an open problem. This problem occurs due to the complex non-linear behavior and redundancies of continuum robots.

The constant curvature assumption is a widely accepted approach and is used to simplify the modeling of continuum robots. It states the following: if each section is optimally constructed, then each section will bend such that the curvature and torsion are approximately constant [11]. Together with the assumption that robot construction

does not allow torsion, the constant curvature assumption allows describing the section backbone as a circle arc. In its turn, the end-effector of the arc can be described by a virtual robot comprising one rigid link with variable length and two to five revolution joints [12]. This significantly reduces the kinematic model parameters.

The constant curvature assumption is used to define the forward and inverse kinematics of continuum robots comprising non-extensible [11–13], extensible [14,15], and telescopic sections [16,17]. The most common way to solve the inverse kinematics of constant curvature multisection continuum robots is the usage of Jacobian-based methods [12,14]. Other options are geometric [13,18] and analytical (not appliable to multisection robots comprising non-extensible sections) methods [15], learning-based [19] methods, and FABRIK-based (forward- and backward-reaching inverse kinematics) algorithms [20,21].

However, a section of a real continuum is not optimally constructed, and thus the constant curvature assumption does not properly model the backbone of the bending section, especially when the external forces are non-negligible or the section is designed in such a way that constant curvature is not appliable. A more realistic way to model continuum robots is to use variable curvature. The variable-curvature kinematic model can be described by Euler spirals [22,23], Pythagorean hodograph curves [24], mode shape functions [25], and piecewise constant curvature assumption [26]. Piecewise constant curvature assumption describes the deformation of a single section with a finite number of serially connected circular arcs—subsections. The piecewise constant curvature assumption is used to model more realistic kinematics of conically shaped continuum robots.

Nowadays, several approaches are employed to solve the inverse kinematics of multisection continuum robots with variable curvature. The Jacobian-based approach is a commonly used approach to find inverse kinematics solutions [26]. It uses the minimum-norm and minimax iterative algorithms to solve the inverse kinematics problem, e.g., Newton's method. The Jacobian-based methods are accurate and capable of working in real time. However, they suffer from a high computational cost, complex matrix calculations, singularity problems, and the inability to find a solution in some cases.

Model-free approaches are used to solve inverse kinematics and control continuum robots without an a priori known robot model. An example of such an approach is represented by neural networks [27,28]. They create models based on data from the real robot or accurate models to predict the inverse kinematics solution. Such models allow fast computation of inverse kinematics without knowing physics-based models in advance. The trained models achieve both a low error rate and low latency. However, the learning-based approach is badly scalable and requires training a new model for each new robot design.

Model-free approaches are also used for feedback control of continuum robots. They show better accuracy than constant curvature models because they can implement the control policies directly from the task space in the actuation space, avoiding model mismatches. Model-free approaches that are used to control continuum robots are based on: Fuzzy Logic [29], recurrent neural networks [30], and reinforcement learning [31].

The next approach is based on bioinspired algorithms such as particle swarm optimization [32–34], genetic algorithms [33,34], and artificial bee colony algorithms [33]. These algorithms minimize the error between the current and desired end-effector position and orientation by changing their configuration variables. All the mentioned algorithms successfully solve inverse kinematics in real time. However, current studies are limited to three-section continuum robots; in this regard, the scalability of bioinspired algorithms is unknown.

In this paper, we present FABRIKx—a fast and reliable algorithm for solving the inverse kinematics of variable-curvature multisection continuum robots. We chose FABRIK [35] as the base algorithm due to its singularity-free nature demonstrating good results for constant curvature continuum robots [20,36]. We combine both tangent [20] and chord [21] approaches with the piecewise constant curvature assumption. To do that, a section of the continuum robot is simplified to one virtual revolution joint and two virtual links with variable lengths. The links are tangent to the beginning and the end of the section.

A joint is placed at their intersection. Tangent links are used to define the preliminary pose of the robot during the forward-reaching stage. During the backward-reaching stage, the chords are used to define section configuration parameters by single-section inverse kinematics. Those parameters are used to redefine link lengths and robot pose.

Our study describes several ways to obtain an inverse kinematics solution for a single bending section that is described by the piecewise constant curvature assumption. To prove FABRIKx's effectiveness, we compare it with the Jacobian-based algorithm and FABRIKc-based algorithm in simulation for different robots. An overview of the whole algorithm is presented in Figure 1.

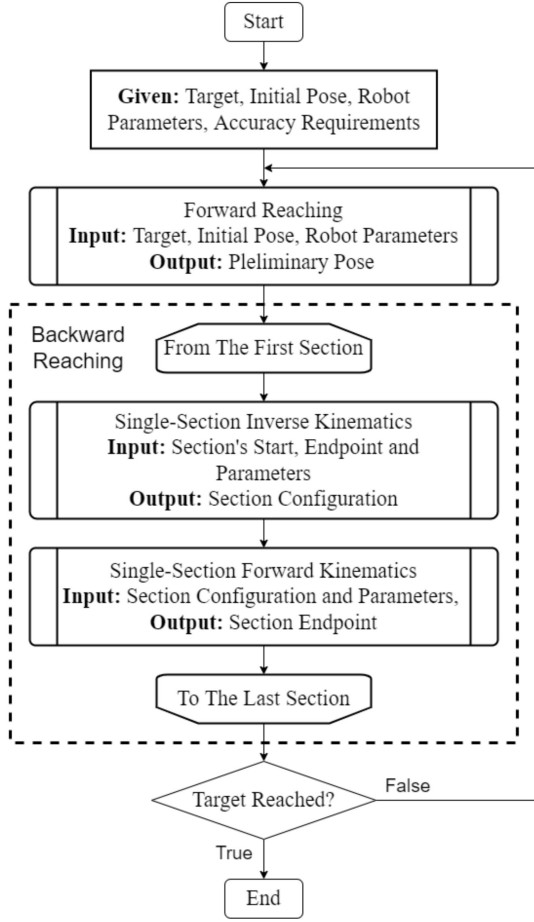

**Figure 1.** A flowchart of the FABRIKx algorithm.

## 2. Forward Kinematics

To describe piecewise constant curvature forward kinematics, we use several assumptions [26]:

1.  Continuum robots consist of $N$ independent, consecutive sections (Figure 2a). Two bending sections have the same tangent vector at the point of their connection.
2.  Each section can be divided into $M$ subsections (Figure 2b). All subsections bend simultaneously when the section bends. The angle between the first and the last tangent is the bending angle of the section. Two subsections have the same tangent vector at the point of their connection. The robot design excludes section torsion.
3.  The constant curvature assumption describes subsection shape (Figure 2c). Therefore, the end-effector of a subsection can be defined using a virtual rigid robot (Figure 2d).

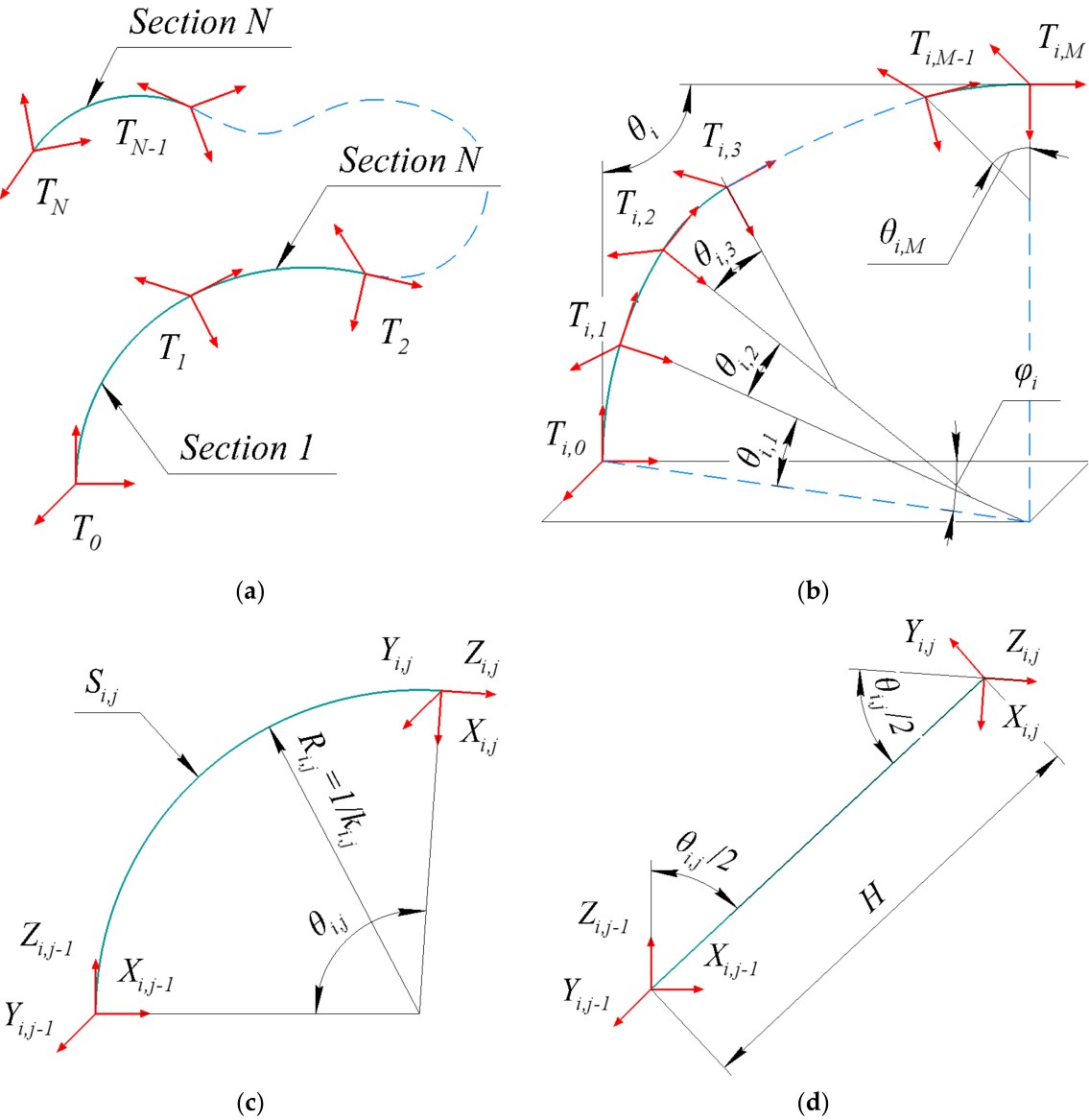

**Figure 2.** Kinematics structure of a variable-curvature multisection continuum robot. (**a**) General robot kinematics. (**b**) Kinematics of the *i*-th section. (**c**) Kinematics of the *j*-th subsection. (**d**) A virtual rigid robot that describes a subsection.

According to assumption 3, a subsection is a circular arc that can be described by a virtual rigid robot. Transformation matrix $B_{i,j}$ describing the bending of the *j*-th subsection of the *i*-th section is defined as follows:

$$B_{i,j} = \begin{pmatrix} \cos(\theta_{i,j}) & 0 & \sin(\theta_{i,j}) & S_{i,j} \cdot \frac{1-\cos(\theta_{i,j})}{\theta_{i,j}} \\ 0 & 1 & 0 & 0 \\ -\sin(\theta_{i,j}) & 0 & \cos(\theta_{i,j}) & S_{i,j} \cdot \frac{\sin(\theta_{i,j})}{\theta_{i,j}} \\ 0 & 0 & 0 & 1 \end{pmatrix} \quad (1)$$

where $S_{i,j}$ is the subsection length, and $\theta_{i,j}$ is the subsection bending angle that is defined as follows:

$$\theta_{i,j} = \frac{w_{i,j} \cdot \theta_i}{\sum_{k=1}^{M} w_{i,k}} \quad (2)$$

where $\theta_i$ is the section bending angle, and $w_{i,j}$ is the subsection weight. Subsection weights are defined during robot construction.

According to assumption 2, a section is a sequence of subsections without torsion between them. Therefore, subsections of a single section are bent in the same plane. Transformation matrix $B_i$ that describes the bending of *i*-th section is defined as follows:

$$B_i = T_Z(\varphi_i) \prod_{j=1}^{M} B_{i,j} \tag{3}$$

where $M$ is the number of subsections, $T_Z(\varphi)$ is the rotation around the *z*-axis at the section rotation angle $\varphi$, and $T_{i,0}$ is the section base.

According to assumption 1, a continuum robot is a sequence of sections. The position and orientation of the robot end-effector $T_N$ is defined as follows:

$$T_N = T_0 \prod_{i=1}^{N} B_i \tag{4}$$

where $T_0$ is the robot base, and $N$ is the number of sections.

### 3. Inverse Kinematics

*3.1. Single-Section Inverse Kinematics*

The single-section inverse kinematics defines a single possible robot configuration that satisfies the end-effector position $P_T$. The single-section inverse kinematics requires the following input data: target point $P_T$, subsection lengths $S$, section weights $w$, and section base frame $T_0$. The output data are configuration variables, such as the bending angle $\theta$ and the rotation angle $\varphi$.

The rotation angle $\varphi$ of the section is defined using the $P_T^X$ and $P_T^Y$ components of the target point $P_T$ in the section base frame $T_0$:

$$P = T_0^{-1} \cdot P_T \tag{5}$$

$$\varphi = atan2(P_Y, P_X) \tag{6}$$

In this work, we use three different approaches to define the bending angle $\theta$.

#### 3.1.1. FABRIKc-Based Method

This approach is inherited from the FABRIKc algorithm [20]. Here, the bending angle is defined as the angle between section tangents:

$$\theta = \text{acos}(Z_{Base} \cdot Z_{End}) \tag{7}$$

where $Z_{Base}$ is the *z*-axis of the section base, and $Z_{End}$ is the *z*-axis of the section end.

#### 3.1.2. Iterative Method

The bending angle can be found through the chord angle. For constant curvature sections, the ratio of the bending angle $\theta$ to the chord angle $\alpha$ is known. It is constant and the same for all constant curvature sections. However, for the variable-curvature section, this ratio depends on subsection lengths, weights, and the current bending angle. Examples of the ratio between the bending angle $\theta$ and the chord angle $\alpha$ for the three different sections (parameters presented in Table 1) are shown in Figure 3.

**Table 1.** Section parameters.

| Section Color | Subsection Lengths, mm | Subsection Weights |
| --- | --- | --- |
| Red | [10, 20, 40] | [3, 2, 1] |
| Blue | [10, 20, 40] | [1, 2, 3] |
| Yellow | [40, 20, 10] | [3, 1, 1] |

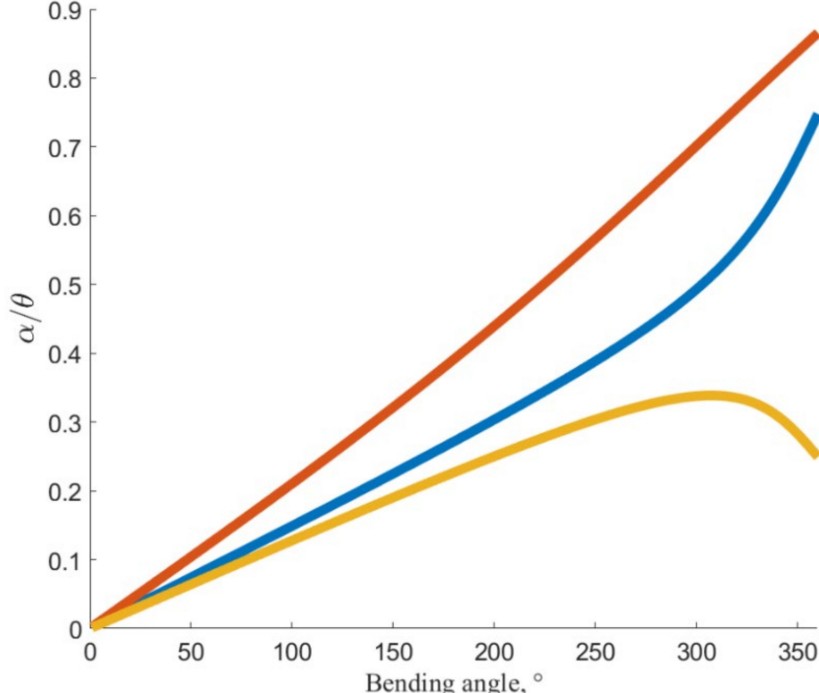

**Figure 3.** The ratio between the bending angle $\theta$ and the chord angle $\alpha$ for different variable-curvature sections. The colors are explained in Table 1.

An iterative method of finding the bending angle $\theta$ using the chord angle $\alpha$ is the following:

1. Define chord angle $\alpha$:

$$\alpha = \mathrm{acos}\left(\frac{Z_{Base} \cdot P}{|P|}\right) \tag{8}$$

2. Assume that bending angle $\theta'$ is equal to chord angle $\alpha$:

$$\theta' = \alpha \tag{9}$$

3. Define section endpoint $P'$ using forward kinematics (1)–(3) for bending angle $\theta'$ and rotation angle $\varphi$.
4. Define chord angle $\alpha'$ for point $P'$ by (8).
5. If the difference between angles $\alpha$ and $\alpha\prime$ is lower than the tolerance $\epsilon$, then $\theta'$ is a solution. Otherwise, redefine $\theta'$ as follows:

$$\theta' = \alpha \cdot \theta' / \alpha' \tag{10}$$

Then, go to step 3.

A flowchart of the single-section iterative inverse kinematics is presented in Figure 4.

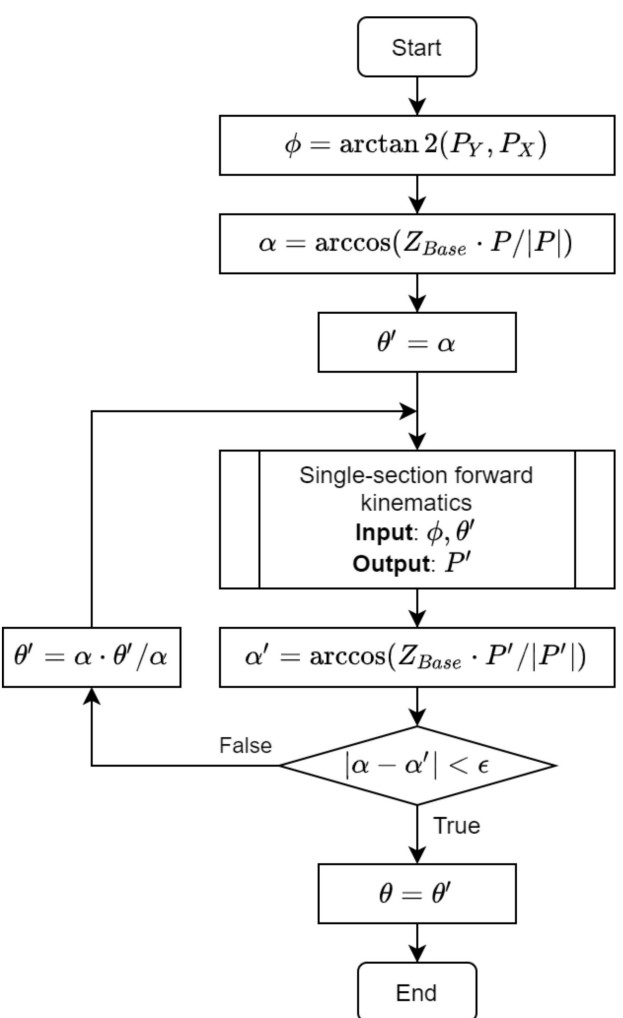

**Figure 4.** A flowchart of the single-section iterative inverse kinematics algorithm.

3.1.3. Approximation

A simpler way to get the bending angle from the chord angle is to use approximation. To do this, a set of bending angles and the corresponding chord angles must be obtained in advance. Then, a function properly describing this data has to be selected. This approach is faster than the iterative one; however, its accuracy depends on approximation quality.

*3.2. Multisection Inverse Kinematics*

The inverse kinematics algorithm of a multisection robot searches for a robot configuration (set of bending and rotation angles) that can reach a target position $P_T$ with a linear tolerance of $\varepsilon_L$ and a target orientation $Z_T$ with an angular tolerance of $\varepsilon_A$.

The algorithm starts with some initial pose, where all $(2N + 1)$ keypoints are known. The keypoints are the beginning of the robot, the ending of each section, and tangent intersection points. Lengths $L$ of tangent links are defined based on keypoints:

$$L_{2i-1} = |P_{2i}P_{2i-1}| \tag{11}$$

$$L_{2i} = |P_{2i}P_{2i+1}| \tag{12}$$

where $i = 1 \dots N$ is the section number.

Each iteration of the algorithm comprises forward-reaching and backward-reaching stages. An example of an iteration of inverse kinematics for a three-section continuum robot is reflected in Figure 5.

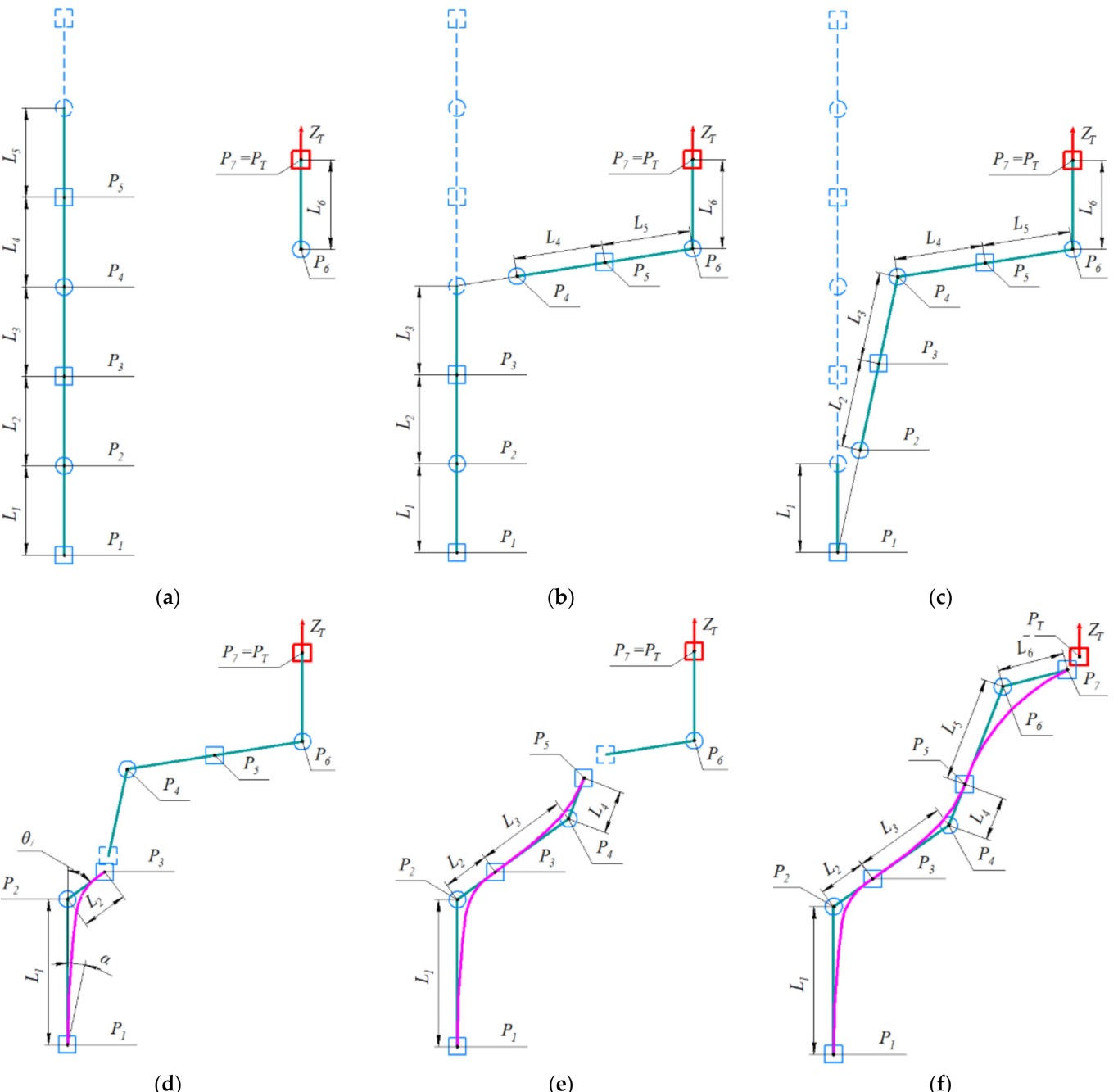

**Figure 5.** A showcase of the FABRIKx iterative algorithm for a three-section continuum robot. (**a**–**c**) Forward-reaching stage. (**d**–**f**) Backward-reaching stage.

### 3.2.1. Forward Reaching

During the forward-reaching stage, the algorithm processes the robot as if it comprises only rigid links connected by spherical joints with no restrictions on the rotation angles. These links are section tangents.

In the first step the point $P_N$ is moved to the target position $P_T$:

$$P_N = P_T \tag{13}$$

Then, the value of the tangent vector $V$ is set as follows:

$$V = -Z_T \tag{14}$$

If $Z_T$ is not specified, then:

$$V = P_N P_{N-1} / |P_N P_{N-1}| \tag{15}$$

Then, all remaining keypoints are redefined starting from the last section $N$ to the first section. In order to do that, the following steps are taken. A tangent intersection point is defined as follows:

$$P_{2(N-i)} = P_{2(N-i)+1} + V \cdot L_{2(N-i)} \tag{16}$$

The tangent vector is defined as follows:

$$V = \begin{cases} P_{2(N-i)} P_{2(N-i-1)} / \left| P_{2(N-i)} P_{2(N-i-1)} \right|, & if \;\; i \neq N-1 \\ -Z_0, & otherwise \end{cases} \tag{17}$$

The endpoint of the section is defined as:

$$P_{2(N-i)-1} = P_{2(N-i)} + V \cdot L_{2(N-i)-1} \tag{18}$$

where $i = 0 \ldots N - 1$.

### 3.2.2. Backward Reaching

Next comes the backward-reaching stage. During this stage, the algorithm defines the configuration parameters $(\theta, \varphi)$ of each section from the first to the last section $i = 1 \ldots N$ using single-section inverse kinematics for point $P_{2i+1}$ and base $T_{i-1}$. To implement joint limit avoidance, the bending angle $\theta$ cannot be larger than the maximum bending angle $\theta_{max}$:

$$\theta_i = \begin{cases} \theta_i, & if \;\; \theta_i < \theta_{max} \\ \theta_{max}, & otherwise \end{cases} \tag{19}$$

Then, forward kinematics Equations (1)–(4) are used to redefine each section endpoint $P_{2i+1}$.

Next, the tangent lengths are updated using the law of sines (see Figure 6):

$$L_{2i-1} = |P_{2i-1} P_{2i+1}| \cdot \sin(\theta_i - \alpha) / \sin(\pi - \theta_i) \tag{20}$$

$$L_{2i} = |P_{2i-1} P_{2i+1}| \cdot \sin(\alpha) / \sin(\pi - \theta_i) \tag{21}$$

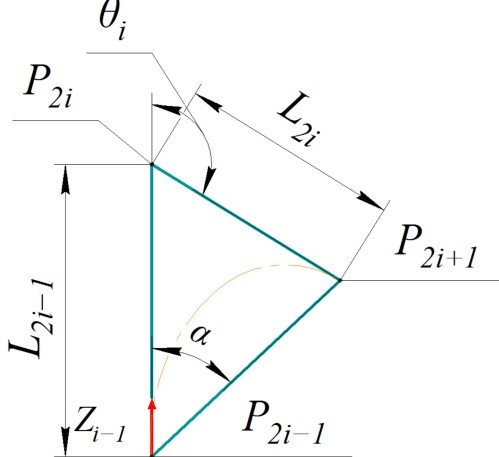

**Figure 6.** Angles of a single section.

For the bending angle $\theta = 0$:

$$L_{2i-1} = L_{2i} = |P_{2i-1} P_{2i+1}| / 2 \tag{22}$$

The tangent intersection point is updated as follows:

$$P_{2i} = Z_{i-1} \cdot L_{2i-1} + P_{2i-1} \tag{23}$$

When the backward-reaching stage is finished, the linear and angular errors are calculated. If the linear or angular error is bigger than the tolerance, then the algorithm performs another iteration starting from forward reaching. Otherwise, the received bending and rotation angles are the final solution of the inverse kinematics. A flowchart of the FABRIKx algorithm is presented in Figure 7.

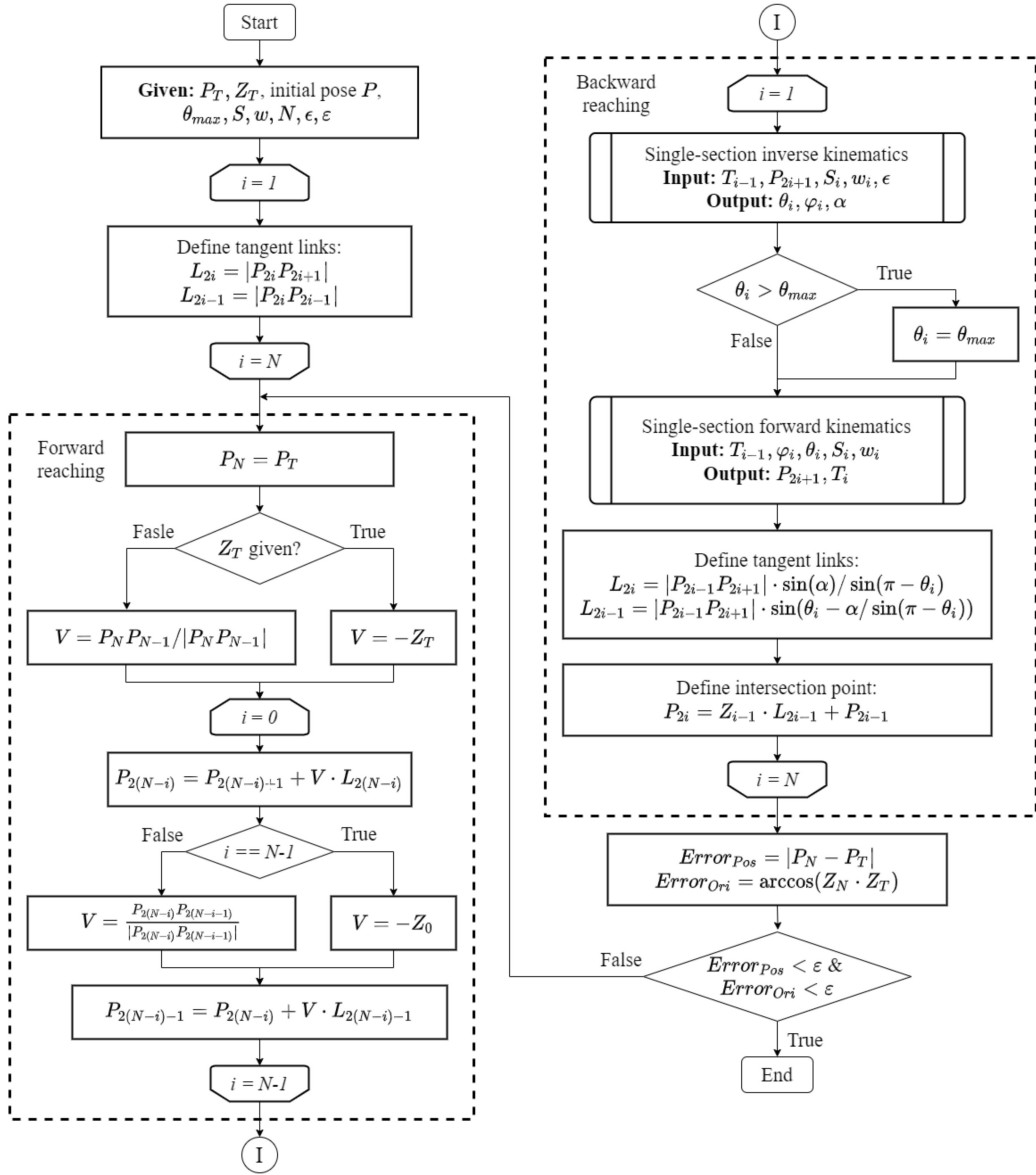

**Figure 7.** A detailed flowchart of the FABRIKx algorithm.

### 3.3. Jacobian-Based Inverse Kinematics

The Jacobian-based approach is the most common way to solve the inverse kinematics problem for both continuum and traditional rigid robots. In this study, we use Newton's method. According to this method, configuration parameters are updated as follows:

$$x_{k+1} = x_k + \left( J(x_k)^T \cdot J(x_k) + W \right)^{-1} \cdot J(x_k) \cdot (G - F(x_k)) \tag{24}$$

where $x_k$ and $x_{k+1}$ are the current and next robot configuration parameters, respectively, $J(x_k)$ is the Jacobian matrix with the parameters $x_k$, $F(x_k)$ is the current position and the current angular error for the parameters $x_k$, $W$ is the damping factor, and $G$ is a vector consisting of the coordinates of the target point and desired angle between the current orientation vector and the desired orientation vector (equal to 0):

$$G = (X_T \; Y_T \; Z_T \; 0)^T \tag{25}$$

The current position $F(x_k)$ is defined by forward kinematics and the angle between the target and the current vector:

$$F(x_k) = \begin{pmatrix} P_{N,x} \\ P_{N,y} \\ P_{N,z} \\ \arccos(Z_T \cdot Z_Q(x_k)) \end{pmatrix} \tag{26}$$

## 4. Simulation

In order to prove the effectiveness of the FABRIKx algorithm, we tested it on a wide variety of robot designs (Table 2), including the following:

1.  Three-section robot with three subsections per section.
2.  Three-section robot with five subsections per section.
3.  Three-section robot with seven subsections per section.
4.  Three-section robot with nine subsections per section.
5.  Five-section robot with three subsections per section.
6.  Seven-section robot with three subsections per section.

Each section has two DOFs, including bending (limited by 100°) and rotation. Each section has a unique workspace (several examples are shown in Figure 8). The initial pose of every robot implies that all bending and rotation angles are set to 0.

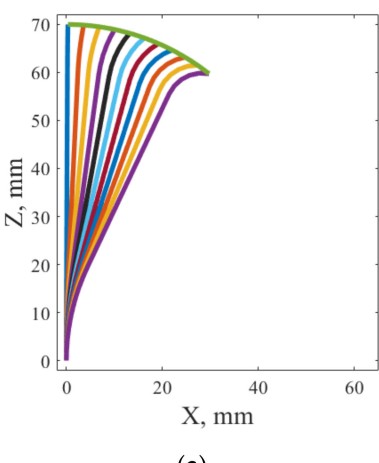
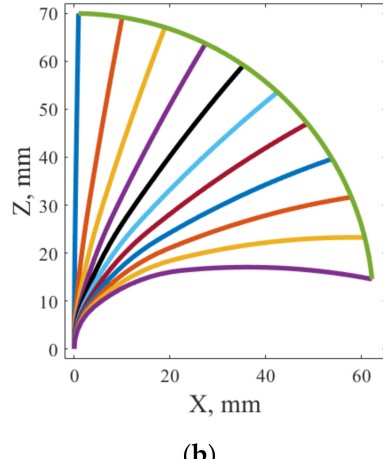
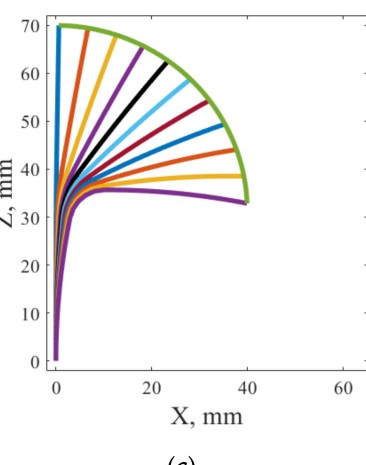

(**a**)                        (**b**)                        (**c**)

**Figure 8.** Sections of Robot 1: (**a**) Section 1; (**b**) Section 2; (**c**) Section 3. Green curves are the section workspaces, while the curves of other colors represent section bodies for bending angles from 0 to 100° with a step of 10° (clockwise in the figure).

FABRIKx was used in two variations to solve single-section inverse kinematics, namely an iterative method with accuracy $\epsilon = 0.006°$ (I) and a 10-degree polynomial approximation (P). Both the Jacobian-based algorithm and the FABRIKc-based algorithm are used for comparison.

We consider that the algorithm reaches the target if the linear error is less than 10 μm and the angular error is less than 0.01°. The algorithm running time was limited to 30 ms.

We note that the running time was increased up to 0.5 s for Robots 5 and 6 using the Jacobian-based algorithm. This case is labeled by the prefix (S).

We used $10^5$ target positions and orientations as input of the inverse kinematics algorithms for each robot. For this sample size, the sampling error reaches 0.41% for a 99% confidence level. Forward kinematics provides each target using random input. Using forward kinematics ensures that all targets have at least one solution of inverse kinematics.

**Table 2.** Robot parameters.

|  | Section | Subsection Lengths, mm | Subsection Weights |
|---|---|---|---|
| Robot 1 (3 × 3) | 1 | [20, 40, 10] | [1, 0.001, 3] |
|  | 2 | [10, 20, 40] | [3, 2, 1] |
|  | 3 | [30, 10, 30] | [1, 7, 1] |
| Robot 2 (3 × 5) | 1 | [20, 20, 40, 10, 10] | [1, 1, 0.001 3, 4] |
|  | 2 | [10, 10, 20, 40, 20] | [4, 3, 2, 1, 0.001] |
|  | 3 | [15, 30, 10, 30, 15] | [0.001, 1, 7, 1, 3] |
| Robot 3 (3 × 7) | 1 | [30, 20, 20, 40, 10, 10, 10] | [4, 1, 1, 0.001, 3, 4, 1] |
|  | 2 | [30, 10, 10, 20, 40, 20, 10] | [1, 4, 3, 2, 1, 0.001, 4] |
|  | 3 | [20, 15, 30, 10, 30, 15, 20] | [1, 0.001, 1, 7, 1, 3, 1] |
| Robot 4 (3 × 9) | 1 | [10, 30, 20, 20, 40, 10, 10, 10, 30] | [2, 4, 1, 1, 0.001, 3, 4, 1, 3] |
|  | 2 | [30, 30, 10, 10, 20, 40, 20, 10, 10] | [4, 1, 4, 3, 2, 1, 0.001, 4, 1] |
|  | 3 | [20, 20, 15, 30, 10, 30, 15, 20, 20] | [4, 1, 0.001, 1, 7, 1, 3, 1, 4] |
| Robot 5 (5 × 3) | 1 | [20, 40, 10] | [1, 0.001, 3] |
|  | 2 | [10, 20, 40] | [3, 2, 1] |
|  | 3 | [30, 10, 30] | [1, 7, 1] |
|  | 4 | [10, 50, 10] | [0.001, 1, 0.001] |
|  | 5 | [10, 50, 10] | [3, 1, 3] |
| Robot 6 (7 × 3) | 1 | [20, 40, 10] | [1, 0.001, 3] |
|  | 2 | [10, 20, 40] | [3, 2, 1] |
|  | 3 | [30, 10, 30] | [1, 7, 1] |
|  | 4 | [10, 50, 10] | [0.001, 1, 0.001] |
|  | 5 | [10, 50, 10] | [3, 1, 3] |
|  | 6 | [20, 40, 10] | [3, 0.001, 1] |
|  | 7 | [10, 40, 20] | [3, 0.001, 1] |

## 5. Results and Discussion

Simulations were carried out using MATLAB 2021b on Windows 10 with an Intel Core i7-4790K 4.00 GHz CPU and 16 GB RAM. The performance of the algorithms was evaluated by success rate and operating time. The obtained simulation results are presented in Table 3.

The results verify the effectiveness of the proposed algorithm. Both FABRIKx variants show a high success rate (from 58.9% to 93.8%). The difference between the success rates of the FABRIKx variants is small (lower than 2%) and appears because of the higher speed of FABRIKx (P). In general, the FABRIKx algorithm outperforms the FABRIKc-based algorithm by 1–15% and the Jacobian-based algorithm by 0–60% in terms of success rate.

The differences in the success rates of Robots 1–4 show that the success rate of FABRIKx decreases with an increasing number of subsections. We assume that using 2-degree polynomial interpolation, the success rate will reach ~50% at 16 subsections of the three-section continuum robot. This case demonstrates that FABRIKx is scalable in subsection number. At the same time, the Jacobian-based algorithm improves its success rate with the increase of the subsection number. FABRIKx (P) and Jacobian-based algorithms become equal in terms of success rate at nine subsections of the three-section continuum robot.

Differences in the success rates between Robots 1, 5, and 6 show that the number of sections significantly affects the success rate of all inverse kinematics algorithms. The success rates of the FABRIKx, FABRIKc-based, and Jacobian-based algorithms dropped to

approximately 60%, 53%, and 0%, respectively. Increasing the running-time limit improves the results of the Jacobian-based algorithm, but it is still significantly low (5%) and unable to operate in real time. This proves that the proposed algorithm has better scalability in the number of sections.

The difference in success rate (up to 15%) between the FABRIKx and FABRIKc-based algorithms proves that using chords in the backward-reaching stage is more effective than using tangents. The FABRIKc-based algorithm shows the same or up to 1.5 times slower solution speed compared to the FABRIKx (P) algorithm.

**Table 3.** Simulation results.

| | Algorithm | Success Rate, % | Solution Time, ms | | Iteration | |
|---|---|---|---|---|---|---|
| | | | Mean | Median | Mean | Median |
| Robot 1 (3 × 3) | FABRIKx (I) | 92.8 | 4.5 | 3.2 | 16 | 11 |
| | FABRIKx (P) | 93.8 | 3.3 | 2.1 | 17 | 11 |
| | FABRIKc | 78.5 | 4.4 | 2.3 | 29 | 14 |
| | Jacobian | 81.3 | 8.1 | 5.9 | 34 | 25 |
| Robot 2 (3 × 5) | FABRIKx (I) | 92.4 | 5.3 | 4.0 | 17 | 12 |
| | FABRIKx (P) | 93.6 | 3.6 | 2.4 | 18 | 12 |
| | FABRIKc | 79.5 | 4.3 | 2.2 | 27 | 13 |
| | Jacobian | 83.7 | 7.9 | 6.0 | 32 | 24 |
| Robot 3 (3 × 7) | FABRIKx (I) | 86.7 | 6.4 | 4.7 | 19 | 13 |
| | FABRIKx (P) | 88.5 | 4.2 | 2.8 | 20 | 14 |
| | FABRIKc | 75.8 | 4.9 | 2.6 | 30 | 15 |
| | Jacobian | 84.4 | 7.3 | 5.1 | 28 | 19 |
| Robot 4 (3 × 9) | FABRIKx (I) | 82.8 | 6.7 | 4.9 | 19 | 14 |
| | FABRIKx (P) | 84.9 | 4.4 | 2.9 | 21 | 14 |
| | FABRIKc | 73.4 | 5.0 | 2.7 | 30 | 15 |
| | Jacobian | 84.9 | 7.4 | 5.2 | 27 | 19 |
| Robot 5 (5 × 3) | FABRIKx (I) | 69.3 | 5.8 | 4.5 | 12 | 10 |
| | FABRIKx (P) | 70.7 | 4.1 | 3.1 | 13 | 10 |
| | FABRIKc | 70.0 | 4.0 | 3.0 | 16 | 12 |
| | Jacobian | 0.3 | 26.0 | 26.6 | 28 | 28 |
| | Jacobian (S) | 13.3 | 138.7 | 98.4 | 128 | 90 |
| Robot 6 (7 × 3) | FABRIKx (I) | 58.9 | 7.4 | 6.0 | 12 | 10 |
| | FABRIKx (P) | 60.4 | 5.4 | 4.1 | 13 | 10 |
| | FABRIKc | 53.1 | 8.2 | 6.2 | 24 | 18 |
| | Jacobian | 0 | N/A | N/A | N/A | N/A |
| | Jacobian (S) | 5.2 | 401.3 | 411.0 | 60 | 61 |

The solution time shows that FABRIKx can operate in real time. FABRIKx (P) outperforms the Jacobian-based algorithm by 1.5–2 times in solution time. FABRIKx (I) has the same solution time or 1.5 times faster than the Jacobian-based algorithm. The difference between the solution time for both variants of FABRIKx shows that the algorithm can be further improved by using more accurate and/or faster single-section forward and inverse kinematics algorithms. For example, a neural network trained on real robot data or an accurate model can be used to achieve accurate and fast solutions [37]. Another possible option is to replace two sections of a robot with two virtual tangent links. This is possible because an analytical solution is achievable for the given position and orientation of the endpoint.

The achieved results demonstrate that the proposed FABRIKx algorithm can improve autonomy, motion planning, trajectory optimization, and obstacle avoidance of multisection continuum robots with variable curvature by providing fast and accurate inverse kinematics solutions. Having inherited the nature of the FABRIK algorithm, the proposed FABRIKx algorithm possesses a wide variety of advantages including simple implementation, low



latency, lack of singularity, an easy-to-modify approach, and scalability in both the number sections and subsections.

## 6. Conclusions

The paper proposes the FABRIKx algorithm which can be used for solving inverse kinematics of a multisection continuum robot of variable curvature. In the algorithm, variable curvature is approximated by a piecewise constant curvature assumption. The proposed FABRIKx combines both tangent and chord approaches to solve the inverse kinematics problem. In particular, tangents are used to define the preliminary positions of the robot during the forward reaching stage while chords are used to define the final pose of the robot by single-section forward and inverse kinematics during the backward reaching stage. Additionally, we present a way to solve a single-section inverse kinematics problem as a special case.

FABRIKx is able to identify solutions for 58.9% to 93.8% of scenarios. For the majority of studied scenarios, the algorithm outperforms the FABRIKc-based algorithm by 1–15% and the Jacobian-based algorithm by 0–60% in terms of success rate. Also, the key advantages of the algorithm are the scalability in both the number of sections and subsections, and it does not suffer from singularity problems. In addition to that, FABRIKx can operate in real-time and works 1.5 to 2 times faster than the Jacobian-based algorithm and up to 1.5 times faster than the FABRIKc-based algorithm. The algorithm allows further modifications by using more accurate and faster single-section forward and inverse kinematics if such information is available for specific scenarios. The proposed algorithm can improve autonomy, motion planning, trajectory optimization, and obstacle avoidance of multisection continuum robots with variable curvature.

**Author Contributions:** Conceptualization, D.K.; methodology, D.K.; software, D.K. and V.D.; validation, D.K.; formal analysis, D.K., O.G. and V.D.; investigation, D.K.; resources, O.G. and V.D.; data curation, D.K.; writing—original draft preparation, D.K. and V.D.; writing—review and editing, V.D.; visualization, V.D.; supervision, O.G.; project administration, O.G.; funding acquisition, D.K., O.G. and V.D. All authors have read and agreed to the published version of the manuscript.

**Funding:** This work was supported by the Russian Foundation for Basic Research no. 20-38-90143 and the Russian Federation Governmental Program 'Nauka' no. FFSWW-2020-0014.

**Data Availability Statement:** Not applicable.

**Conflicts of Interest:** The authors declare no conflict of interest.

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
