# Peer review of "FABRIKx: Tackling the Inverse Kinematics Problem of Continuum Robots with Variable Curvature"

_robotics, doi:10.3390/robotics11060128_

Round 1
Reviewer 1 Report
The authors present an algorithm, FABRIK-x, that uses a piecewise constant curvature assumption to solve the inverse kinematics problem of a multisection continuum robot with variable curvature. They present the forward and inverse kinematics model for this algorithm and some results from modelling work that bench-marks it’s computation efficiency and accuracy against other existing algorithms. The theory is well presented, and the results show good improvements over existing algorithms. However, the manuscript has several areas which should be improved before publication.
Major comments
Section 5 is both results and discussion, however the discussion is largely absent, with no discussion of the results in the context of other work and no effort to elude on questions which arise. For example, in lines 257 to 262 – why do the authors think the number of sections significantly affects success rate? How were singularity problems, identified in the introduction section, addressed in the current work?
Minor comments
A figure (e.g. flowchart) with an overview of the work would be useful before the forwards kinematics section. Since several algorithms are used, it is difficult to discern how each algorithm or step fits into each method using only the title structure within each section. Alternatively, sections 2 and 3 should start with an introductory paragraph.
No legend in Fig 2 – what variable curvatures do the colors represent?
Lines 251 to 256 – this analysis seems unwarranted given only 2 data points are available (Robot 1 and Robot 2). Instead of projecting performance using 2 data points, it would be more prudent to gather several more data points and present a plot (e.g. Robotx (3 x 7), Roboty (3 x 9), etc...)
The manuscript has many syntax errors. While most errors are minor, others make it difficult to understand the text. Running the manuscript through a word processor will detect most or all of these errors.
Author Response
Dear Reviewer,
Thank you for your comments. We very much appreciate your suggestions, which have been very helpful in improving the manuscript. Below, we present detailed answers to the comments received:
Comment 1: Section 5 is both results and discussion, however the discussion is largely absent, with no discussion of the results in the context of other work and no effort to elude on questions which arise. For example, in lines 257 to 262 – why do the authors think the number of sections significantly affects success rate? How were singularity problems, identified in the introduction section, addressed in the current work?
Answer 1: Results and discussion sections extended. Simulation extended by Robots 3 (3x7), Robot 4 (3x9) and Robot 6 (7x3). We suppose extended data is enough to make a conclusion about the effect of increasing sections or subsections number at the success rate. FABRIK is a singularity-free algorithm and FABRIKx inherits this feature.
Comment 2: A figure (e.g. flowchart) with an overview of the work would be useful before the forwards kinematics section. Since several algorithms are used, it is difficult to discern how each algorithm or step fits into each method using only the title structure within each section. Alternatively, sections 2 and 3 should start with an introductory paragraph.
Answer 2: We added a short flowchart (Figure 1) of the algorithm in the introduction.
Comment 3: No legend in Fig 2 – what variable curvatures do the colors represent?
Answer 3: Legend added to Figure 3 (ex Figure 2). Curves are ratio between bending angle and chord angle for 3 different variable curvature sections.
Comment 4: Lines 251 to 256 – this analysis seems unwarranted given only 2 data points are available (Robot 1 and Robot 2). Instead of projecting performance using 2 data points, it would be more prudent to gather several more data points and present a plot (e.g. Robotx (3 x 7), Roboty (3 x 9), etc...).
Answer 4: Simulation extended by Robots 3 (3x7), Robot 4 (3x9) and Robot 6 (7x3).
Comment 5: The manuscript has many syntax errors. While most errors are minor, others make it difficult to understand the text. Running the manuscript through a word processor will detect most or all of these errors.
Answer 5: We fully reviewed and rewrote our manuscript.
Reviewer 2 Report
In this paper, a fast and reliable algorithm is presented to solve the inverse kinematics of the multisection continuum robot of variable curvature. The topic of this paper is worth for being investigated. However, a few issues need to be addressed before this paper can be accepted.
1. The literature review should be updated: for example, a list of the method to solve the kinematics of continuum robots was presented here. However, some other widely-used methods were not included here, which should be added, such as model-less approaches.
one example of state-of-the-art is Ba, W., Dong, X., Mohammad, A., Wang, M., Axinte, D. and Norton, A., 2021. Design and validation of a novel fuzzy-logic-based static feedback controller for tendon-driven continuum robots. IEEE/ASME Transactions on Mechatronics, 26(6), pp.3010-3021.
Further, the latest example of repairing inside complex devices is Dong, X., Wang, M., Mohammad, A., Ba, W., Russo, M., Norton, A., Kell, J. and Axinte, D., 2022. Continuum Robots Collaborate for Safe Manipulation of High-Temperature Flame to Enable Repairs in Challenging Environments. IEEE/ASME Transactions on Mechatronics.
Lilge, S., Barfoot, T.D. and Burgner-Kahrs, J., 2022. Continuum robot state estimation using Gaussian process regression on SE (3). The International Journal of Robotics Research, p.02783649221128843.
Santiago, J.L.C., Walker, I.D. and Godage, I.S., 2015, March. Continuum robots for space applications based on layer-jamming scales with stiffening capability. In 2015 IEEE Aerospace Conference (pp. 1-13). IEEE.
2. In Figure 2, what do different colours represent? A similar problem can be seen in Figure 7?
Overall this is a good problem, presenting a fast and reliable algorithm for solving inverse kinematics. However, the identified issues must be addressed before this paper can be accepted.
Author Response
Dear Reviewer,
Thank you for your comments. We very much appreciate your suggestions, which have been very helpful in improving the manuscript. Below, we present detailed answers to the comments received:
Comment 1: The literature review should be updated: for example, a list of the method to solve the kinematics of continuum robots was presented here. However, some other widely-used methods were not included here, which should be added, such as model-less approaches.
Answer 1: We add [1] as a model-free approach, along with two others (reinforcement learning and recurrent neural networks). However, we don't think that approaches used to feedback control are suitable for a literature review about inverse kinematics algorithms.
[2] and [4] are added as examples of the continuum robots application.
[3] is a shape sensing approach. Article is unrelated to inverse kinematics and can't be used as an example of application, so we didn't add it.
Comment 2: In Figure 2, what do different colours represent? A similar problem can be seen in Figure 7?
Answer 2: Legends added to Figures 3 (ex Figure 2) and 8 (ex Figure 7). Colors in Figure 3 show a ratio between chord angle and bending angle for 3 different sections. Outer curves in Figure 8 is a section workspace, the other curves is a section bodies for different bending angles (from 0 to 100° with step 10°).
- Ba, W., Dong, X., Mohammad, A., Wang, M., Axinte, D. and Norton, A., 2021. Design and validation of a novel fuzzy-logic-based static feedback controller for tendon-driven continuum robots. IEEE/ASME Transactions on Mechatronics, 26(6), pp.3010-3021.
- Dong, X., Wang, M., Mohammad, A., Ba, W., Russo, M., Norton, A., Kell, J. and Axinte, D., 2022. Continuum Robots Collaborate for Safe Manipulation of High-Temperature Flame to Enable Repairs in Challenging Environments. IEEE/ASME Transactions on Mechatronics.
- Lilge, S., Barfoot, T.D. and Burgner-Kahrs, J., 2022. Continuum robot state estimation using Gaussian process regression on SE (3). The International Journal of Robotics Research, p.02783649221128843.
- Santiago, J.L.C., Walker, I.D. and Godage, I.S., 2015, March. Continuum robots for space applications based on layer-jamming scales with stiffening capability. In 2015 IEEE Aerospace Conference (pp. 1-13). IEEE.